# IoT Based Expert System for Diabetes Diagnosis and Insulin Dosage Calculation

**DOI:** 10.3390/healthcare11010012

**Published:** 2022-12-21

**Authors:** Prajoona Valsalan, Najam Ul Hasan, Umer Farooq, Manaf Zghaibeh, Imran Baig

**Affiliations:** 1College of Engineering, Dhofar University, Salalah 211, Oman; 2School of Engineering, University of Sunderland, Sunderland SR6 0AA, UK

**Keywords:** IoT, fuzzy system, diabetes diagnosis and treatment, remote health

## Abstract

High blood glucose levels are the defining characteristic of diabetes. Uncontrolled blood glucose levels in diabetic patients might result in mortality. As a result, there is a dire need to control blood glucose levels by constantly monitoring them and delivering the appropriate amount of insulin. However, insulin consumption is affected by several variables, including age, calorific intake, and body weight. The patient must see the doctor on a regular basis in order to determine the appropriate dose. Nonetheless, hospital facilities are finding it increasingly difficult to treat patients as the number of patients rises; thus, the healthcare industry is searching for an efficient method that can alleviate their burden by assisting patients with chronic conditions through remote patient care. In this work, we have developed an expert system to provide remote treatment for diabetic patients. Our expert system consists of two distinct components: one for the patient and one for the hospital. The sole requirement for the patient will be a wearable device that captures and transmits all relevant data to the cloud. On the hospital side, there should be a system in place to process that data in the cloud. The system employs a fuzzy system to handle data in two stages. A fuzzy system is initially employed to identify whether or not a patient is diabetic. In the second stage, a fuzzy system is utilized to determine the insulin dosage for a diabetic patient. Using sensors and the ESP8266 platform, we have developed a prototype of patient-side hardware. The MATLAB fuzzy toolbox is used for the processing part, which includes fuzzy systems, and the results of the MATLAB analysis are presented in the form of simulation results to demonstrate the accuracy of the proposed system in terms of determining insulin dosage. The results of the simulation using the fuzzy toolbox for the insulin dose of the diabetic patient are significantly close to the amount of dosage prescribed by the endocrinologist.

## 1. Introduction

Remote patient monitoring (RPM) is a medical care delivery system that allows patients to be monitored outside traditional medical settings, such as at home or some other remote area by means of advanced information technology. RPM is quite useful in several aspects, such as day-to-day patient health monitoring, increasing physician capacity to serve patients, lowering healthcare costs, and much more. In recent years, RPM has become extremely significant for medical care professionals, IT specialists, bioinformatics experts, and others. As per the information from the United Nations Population Fund (UNFPA) [1], the number of people aged above 60 will increase to 2 billion by 2050. Added to this, as per a report by the World Health Organization (WHO), there was a shortage of about 7.2 million healthcare workers recorded in 2013, which is estimated to become 12.9 million by 2035. Currently, a good proportion of the elderly population are diagnosed with many age-related health issues like diabetes and cardio- vascular disease. Therefore in a pandemic such as the one caused by COVID-19, RPM is becoming increasingly important in reducing the stress on health services, with particular emphasis on hard-hit regions, as patients with chronic conditions such as diabetes, heart disease, and liver disease can receive in-house medical care. In addition, non-critical patients requiring hospitalization can be followed constantly in real-time. RPM can considerably benefit people with chronic diseases. It also lowers the cost of patient treatment by reducing the number of clinical trips as well as the burden on medical professionals. According to world bank development indicators, Oman spends the equivalent of 2.6 percent of its Gross Domestic Product (GDP) on health care, which is slightly lower than the GCC average. Population growth will result in a 210 percent increase in demand for health care by 2025, with cardiovascular disease treatment accounting for 21 percent of total health care demand. Moreover, this budget is growing at a rapid pace every year and has an impact on different economies around the globe. Oman’s healthcare expenditures are projected to rise by 12.9 percent by the end of 2022, with total spending expected to be USD 4.3 billion, up from USD 3.4 billion in 2016 [2]. One of the reasons for this is the overuse of Oman’s healthcare services by patients and doctors. The clinic receives 5.6 visits per person per year on average, and 70 percent of the tests performed by doctors are unnecessary [3].

Diabetes is a disease that requires continual monitoring and management; thus, it typically necessitates more doctor visits than other conditions. According to the doctors of the American Diabetic Association, diabetic patients must visit a doctor at least once every 3–4 months [2,3]. Thus, diabetes patients are among the most commonly known case studies of RPM. Diabetes is a life-long chronic disease that is recognized worldwide as the leading cause of premature death and disability. Every year, many countries around the world spend billions of dollars on diabetes treatment. Complications from poor diabetes management, many of which are preventable, account for a large portion of these costs. The risk of these complications can be minimized by monitoring daily blood sugar values and maintaining levels close to normal by prescribing the correct dosage of insulin. It requires close monitoring of vital signs and effective communication between the patient and his healthcare professionals. As it is impractical for patients and healthcare professionals to do so daily in normal clinical settings, RPM is a viable answer to this issue. By integrating the wearable sensors with the wireless communication system, it has made the healthcare services go from clinic-centric to patient-centric [4].

In this study, we have developed an expert system to enable RPM for patients with diabetes. The developed expert system can be subdivided into two subsystems as shown in Figure 1. The first subsystem is meant for data acquisition purposes. The patient is equipped with wearable sensors for the acquisition of vitals. These sensors are connected to his smartphone via Bluetooth. An application on the smartphone transmits the sensor readings and patient data to the cloud via the internet. The second subsystem is utilized to diagnose and treat diabetes when data is saved in the cloud. This is a multi-stage system. In the first stage, a fuzzy logic-based program classifies the patient as diabetic or non-diabetic. In the second stage, a fuzzy logic-based program prescribes an insulin intake for the diabetic patient. The outcomes of the program will be visible to the patient as well as the health care practitioner. The health care practitioners can also add their recommendations to the top prescription generated by the system. The major contributions of this paper can be summarized as follows:An expert system has been developed that can be divided into two subsystems, one owned by the patient and the other owned by the medical care unit.The subsystem at the patient end is used to obtain the body vitals of the patient via the implanted devices and transmit them regularly to the second subsystem.The second subsystem has two levels of decision-making, both of which use fuzzy logic-based decision-making processes.The first fuzzy system is used to diagnose whether or not the patient is diabetic and the second is to determine the dose of insulin required for the diabetic patient.

In the case of a healthy person, feedback is provided after the first fuzzy process, but a diabetic patient receives feedback after the second fuzzy process. The rest of the paper is organized as follows; the related work on diabetic treatment through RPM is discussed in Section 2. The system model and architecture of the proposed expert system to diagnose diabetic and the fuzzy logic mechanism to prescribe insulin dosage intake is set out in detail in Section 3. The experimental setup is discussed in Section 4, the results of the simulation are shown in Section 5 and the paper is concluded in Section 6.

## 2. Related Work

The World Health Organization (WHO) predicts that there are currently 451 million diabetic patients worldwide and this number is expected to exceed 693 million by 2045 [3]. In the Sultanate of Oman, as reported by the Ministry of Health, a total of 94,921 cases of diabetes were registered at a national level in 2017, including 6360 new cases in the various governorates of Oman, of which 51.8 percent were women with 3198 cases compared to 2863 men and the numbers are increasing at a significant rate annually [4]. The World Health Organization also predicted that the number of diabetics in Oman would increase by about 217,000 by 2025 [5]. Diabetes is an illness in which the body produces insufficient insulin. Insulin is a hormone that regulates normal blood glucose levels. Treatment of diabetic disease requires regular visits to health care, which is not feasible in the present pandemic situation of COVID-19. Thus, telemedicine is the only option for any medical procedure requiring remote care. Furthermore, this situation is exacerbated by the COVID-19 scenario due to a shortage of medical personnel and healthcare facilities, as the majority of resources are diverted to COVID-19 patients. Therefore, an expert system for diabetes management is much more needed than before. Although RPM-related efforts have been ongoing for a long time, they are now receiving significantly greater attention than in the past. Various smartphone sensor research and reviews for health monitoring and diagnosis along with the cyber threats, use of cloud computing in IoT devices [6] are also being conducted [7]. The study in [8] details an internet of things (IoT)-based health care system including a heart pulse sensor, a body temperature sensor, and a galvanic skin reaction sensor, and demonstrates that the IoT linked with health wearables can eliminate the need to visit hospitals for primary health concerns. In [9], MATLAB is used to simulate a remote healthcare monitoring system that employs an expert system to monitor the health state of a patient via multiple wearable sensors. In this section, we are going to discuss the various efforts that have been recorded related to diabetes management and their benefits, shortcomings, and drawbacks.

While significant effort has already been made in the past to provide medical care services outside the conventional hospital premises, this has become much more relevant with the current pandemic scenario of COVID-19, which can be seen by an increase in the number of publications. Nonetheless, as shown by a number of the most recent survey papers, there is still a great deal of work to be done to implement home delivery of medical diagnosis and care. For example, the authors provided a survey article in [10] to demonstrate the feasibility of using IoTs for mobile health services. The key focus of this article, however, was on using smart bands to monitor vital factors such as heart rate, which can later be shared with family members as well as the doctor via mobile telephony services. A relatively comprehensive survey on the use of IoTs in the provision of healthcare facilities, along with its various applications, use cases and requirements, can be found in [11]. However, providing patient care and facilities at home is not the only emphasis of this proposed study. A survey paper mainly on the collection of vital sign data, for instance, heart rate monitor, body temperature and a galvanic skin response is provided in [8] for elderly patients. In [12], an IoT-based model for monitoring the student health and using vital signs is proposed. The data for detecting the behavioral and biological changes was performed using the machine learning methods. Authors in [13] proposed a health monitoring system based on IoT to check vital signs as well as detect the biological and the behavioral changes of a disabled or elderly person using the smart elderly care technologies. It provides a health monitoring system for the involved medical teams to continuously monitor and assess the behavioral activity as well as the biological parameters, applying sensor technology through the IoT devices and the data analysis is carried out with different machine learning methods. For an improved healthcare quality, an event-driven IoT architecture is proposed in [14] for data analysis of layer parameters using the Complex Event Processing (CEP) method. By analyzing and interpreting the real-time information from the patient using the CEP method, forecasting anomaly, actionable insights, and increased healthcare quality was achieved. The authors in [15] present a wireless body area network (WBAN) focused on IoT health care systems with a major emphasis on improving network services to enhance health care services. The paper introduces, first, a review of the existing IoTs network for health care from the point of view of network architecture, as well as security and power management issues. Later, an architecture was proposed to address the issues under consideration. The above-mentioned works using IoTs for healthcare services are addressed from a general perspective, whereas patients with chronic diseases such as diabetes are not targeted. The theme of work on IoT-based diabetic services can be divided into categories like diabetic diagnosis and treatment. In [16], the authors included a comprehensive and systematic analysis of Machine Learning- and Artificial Intelligence-related techniques used for the detection and self-management of diabetes mellitus. However, in this proposal authors used the traditional methods instead of an IoT-based platform. Authors in [17] proposed an IoT-based system using raspberry pi to monitor diabetic patients. The proposed system acquires body vitals such as the level of glucose, blood pressure, and temperature of the patient, and the information is stored in a cloud server where physicians can access the information. This method takes diabetes into account, however the patient must visit the doctor at the clinic to receive treatment. In [18,19] diabetic diagnosis was performed using adaptive neuro- fuzzy inference system, by integrating the interpolation of fuzzy logic control and the adaptability through a neural network. The proposed system was not based on the IoT platform and focused only on the accuracy of the diabetic diagnosis procedure. In [20], the authors have developed an IoT-based device architecture with an RF communication protocol using sensors for monitoring blood glucose and body temperature in a human-readable and graphical way for both patients and physicians. In addition, the glucose monitoring system focuses on a high degree of energy- efficient diagnosis of the patient’s diabetic condition using continuous glucose monitoring in real time and warnings in case of abnormal conditions, but the treatment stage is overlooked. In [21], the authors assessed the security of data and cyber threats that patients with diabetes can experience while using IoT-based smart glucose monitoring devices. Instead of on the diagnosis or treatment of diabetes conditions, the main emphasis of this work was on the value of protecting the data of diabetic patients. With respect to the management of diabetes, there are different methods of treating diabetics, ranging from healthy diets to the ingestion of oral medications and injectable medications. For example, the authors in [22] proposed a fuzzy expert system for diagnosis of diabetes and insulin dosage based on plasma glucose and Body Mass Index (BMI). The system was not implemented on an IoT platform, the fuzzification and defuzzification of the parameters was based on the standard deviation calculation of all attributes and the simulation results obtained were not performed in MATLAB, due to which the output values calculated were not confirmed with any simulation results. Moreover, one of the most important parameters, the daily carbohydrate intake of the patient, was not considered for calculating the insulin dosage. The authors in [23] proposed a double moving average IT-based glucose monitoring algorithm to avoid the development of diabetic complications in older patients. The work was focused more on the regular monitoring of the diabetes condition rather than prescribing any medicines for it. Working in the same context, in the care of patients with diabetes, the authors in [24] discussed procedures that can be simplified and improved with IoT. They recommended practices such as the use of wearable devices, collecting information on the mental and physical health status of patients and recommending a healthier lifestyle for patients through dietary follow-up and day-to-day activities. In [25], the authors proposed an IoT-based approach to assist in the management of diabetes by monitoring various vitals such as blood pressure, glucose level, calorie intake and physical activity and, as a result, by providing adequate feedback to the person on the management of diabetes using the Intelligent Health Service Approach, risk events can be detected and reported in advance. However, our work is different from the rest of this work in many ways and the details of novelty are described below.

Our work involves both diagnosis and treatment of diabetes.The second part of our work involves calculating the insulin dose based on the health status of the diabetic patient and adjusting the insulin dosage by taking into account the patient’s numerous lifestyle factors.Fuzzy logic-based mechanisms, which are highly recommended for this kind of decision-making process, are used in this work.

## 3. Proposed Expert System

### 3.1. System Architecture

The main objective of this work is to design and implement a system for the prevention, control, management, and treatment of sugar levels in patients with diabetes. For this goal, an expert system was built that can be divided into two major sub-systems. One subsystem operates at the patient’s end and the other operates at the clinic or medical care unit end. Both subsystems collaborate to provide the aforementioned patient care amenities outside of the hospital. First, the patient’s end of subsystem is used to track blood levels and avoid diabetic disease, whereas the second subsystem is used to diagnose the patient’s condition and to administer the medications of the patient.

The subsystem at the patient’s end can be viewed as both a data acquisition and a feedback system. This subsystem is an IoT node that includes a number of sensors, a processing unit, and a communication module. We have utilized the ESP 8266 microcontroller, which is capable of performing Wi-Fi-related tasks and is therefore extensively employed as a Wi-Fi module for data communication. We have also used two sensors- the blood sugar sensor to measure the blood sugar value and the heart rate sensor to measure the heartbeats for the patient. Firstly, it collects the patient’s vitals using various sensors, including a blood sugar sensor and a heart rate sensor embedded in the patient’s body. Then, after processing, it communicates with the second subsystem on a regular basis via the communication module, such as a 4G or 5G network. The amount of data sent depends on how frequently the data is transmitted. One would wonder why we use 4G or 5G if data is not delivered regularly, but the answer is that we also want to transmit data across long distances. When the patient’s vitals are obtained by the medical server at the end of the clinic, a fuzzy logic decision-making process is employed. This fuzzy system for diabetic diagnosis has two inputs, one is blood sugar and the other is heart rate. This subsystem has one output, which is the person’s health state as either healthy or unwell. After a decision has been made on the status of the patient’s well-being, feedback is given to the patient’s subsystem. Even if the patient is well, recommendations can be given to improve his or her well-being, which may be crucial for the prevention of diabetes. On the other hand, if the patient is declared to be diabetic, a second fuzzy logic system for determining insulin dosage, is used for the prescription of the medication. This second fuzzy system has four inputs: (1) health status, (2) blood sugar weight of the patient, (3) daily carbohydrate intake, and (4) one output that is the insulin dose. Once the outputs of both fuzzy systems in terms of health status and insulin amount are obtained, the export system communicates with the subsystem at the patient end. The algorithm for the proposed system can be summarized as: The heart rate and blood sugar values are measured using the heart rate sensor and blood sugar sensor, respectively.If the heart rate and blood sugar is in a normal range, the patient is healthy. In all other possible permutations, the patient is declared unhealthy in the first fuzzy system.If the patient is unhealthy and in a diabetic range, then based on the carbohydrate intake, the weight of the patient and the blood sugar value, the insulin dosage is calculated using the second fuzzy system.

The overall advantages of the proposed structure can be summarized as:The system architecture for the RPM is designed and developed for regular monitoring of diabetic patients.The suggested system’s results can be accessed by doctors and patients at any time, from any location.The proposed system performs diabetic diagnostics as well as prescribes insulin intake for diabetics using Fuzzy logic.

### 3.2. Fuzzy System for Diabetic Diagnosis

Wearable sensors are used to record the vitals of the patient. Every eight hours, the patient’s blood sugar and heart rate sensors are read. The values obtained from the blood sugar sensor and the heart rate sensors are fed as inputs to the fuzzification process as shown in Figure 2. For each input sensor in the process of fuzzification, a membership function must be developed [26]. Since we use two input sensors, for the purpose of generating the membership function, the membership function of the blood sugar input is divided into four ranges: low, average, pre-diabetic (moderate), and diabetic (elevated). Similarly, for the heart rate membership function, three input ranges are considered: Bradycardia (low heartbeat), Normal, and Tachycardia (high heartbeat). This information is given in Table 1 for the different blood sugar and heart rate sensor input ranges.

One of the inputs used for the fuzzification process is the blood sugar values obtained from the blood sugar sensor, the membership function of which is shown in Figure 3. As discussed before, the membership function values for the blood sugar sensor indicate four conditions: Low (0 mmol/L to 3.9 mmol/L), Normal (4.0 mmol/L to 5.4 mmol/L), Prediabetic (5.5 mmol/L to 6.9 mmol/L) and Diabetic (>7.0 mmol/L). The values of the blood sugar membership are as follows:(1)$$\mathrm{Low}=\left\{\begin{array}{cc} 1;\;\;\;\;\;\;\;\;\;\;\; & x\leq 3.9\\ \frac{4\;-\;x}{4\;-\;3.9};\;\;\;\;\;\;\;\;\;\;\; & 3.9<x<4\\ 0;\;\;\;\;\;\;\;\;\;\;\; & x\geq 4 \end{array}\right.$$
$$\mathrm{Normal}=\left\{\begin{array}{cc} 1,\;\;\;\;\;\;\;\;\;\;\;\;\;\;\; & 4\leq x\leq 5.4\;\;\\ \frac{4\;-\;x}{4\;-\;3.9},\;\;\;\;\;\;\;\;\;\;\;\;\;\;\; & 3.9<x<4\\ \frac{5.5\;-\;x}{5.5\;-\;5.4},\;\;\;\;\;\;\;\;\;\;\;\;\;\;\; & 5.4<x<5.5\\ 0,\;\;\;\;\;\;\;\;\;\;\;\;\;\;\; & x\leq 3.9;x\geq 5.5 \end{array}\right.$$
$$\mathrm{Prediabetic}=\left\{\begin{array}{cc} 1,\;\;\;\;\;\;\;\;\;\;\;\;\;\;\; & 5.5\leq x\leq 6.9\\ \frac{x\;-\;5.4}{5.5\;-\;5.4},\;\;\;\;\;\;\;\;\;\;\;\;\;\;\; & 5.4<x<5.5\\ \frac{7.0\;-\;x}{7.0\;-\;6.9},\;\;\;\;\;\;\;\;\;\;\;\;\;\;\; & 6.9<x<7.0\;\\ 0,\;\;\;\;\;\;\;\;\;\;\;\;\;\;\; & x\leq 5.4;x\geq 7.0 \end{array}\right.$$
$$\mathrm{Diabetic}=\left\{\begin{array}{c} 1,\;\;\;\;\;\;\;\;\;\;\;\;\;\;\;\;\;\;\;\;\;\;x\geq 7.0\\ \frac{\;\;x\;-\;6.9}{7.0\;-\;6.9},\;\;\;\;\;\;\;\;\;\;\;\;\;\;\;\;\;\;\;\;\;\;\;6.9<x<7.0\\ 0,\;\;\;\;\;\;\;\;\;\;\;\;\;\;\;\;\;\;\;\;\;\;\;x\leq 6.9 \end{array}\right.$$

Another input selected for fuzzy logic implementation is the heart rate values obtained from the heart rate sensor. Figure 4 displays the heart rate membership function. Membership functions are known to have three different ranges depending on the value of beats per minute (BPM): Low or Bradycardia (<58 BPM), Normal (60–100 BPM) and High or Tachycardia (>102 BPM). The characteristics of the heart rate membership function are described as follows:
$$\mathrm{Bradycardia}=\left\{\begin{array}{c} 1,\;\;\;\;\;\;x\leq 7.0\\ \frac{60\;-\;x}{60\;-\;58},\;\;\;\;\;\;58<x<60\\ 0,\;\;\;\;\;\;x\geq 60 \end{array}\right..$$
$$\mathrm{Normal}=\left\{\begin{array}{cc} 1,\;\;\;\;\;\;\;\;\;\;\; & 60\leq x\leq 100\\ \frac{x\;-\;58}{60\;-\;58},\;\;\;\;\;\;\;\;\;\;\; & 58<x<60\\ \frac{102\;-\;x}{102\;-\;100},\;\;\;\;\;\;\;\;\;\;\; & 100<x<102\;\\ 0,\;\;\;\;\;\;\;\;\;\;\; & x\leq 58;x\geq 102 \end{array}\right.\;$$
$$\mathrm{Tachycardia}=\left\{\begin{array}{c} 1;\;\;\;\;x\geq 102\\ \frac{x\;-\;100}{102\;-\;100};\;\;\;\;100<x<102\\ 0;\;\;\;\;\;x\leq 100 \end{array}\right.\;$$

### 3.3. Rules for Fuzzy Logic

The fuzzy rules are used to infer the output of the fuzzification inputs [27]. Based on the specified membership functions of the two inputs–blood sugar and heart rate–a set of fuzzy rules for the combination of the two input sensors finalizes the fuzzification rule set out in Table 2. The combination of the given ranges of the two input sensors produces a total of twelve rules. The output of these rules is the health status of a person who may be either healthy or unhealthy. In order to understand this better, the rules are further expressed as follows:

RULE 1: IF Blood sugar == Low AND Heart rate == Bradycardia then output health state = Unhealthy

RULE 2: IF Blood sugar == Low AND Heart rate == Normal then output health state = Unhealthy

RULE 3: IF Blood sugar == Low AND Heart rate == Tachycardia then output health state = Unhealthy

RULE 4: IF Blood sugar == Normal AND Heart rate == Bradycardia then output health state = Unhealthy

RULE 5: IF Blood sugar == Normal AND Heart rate == Normal then output health state = Healthy

RULE 6: IF Blood sugar == Normal AND Heart rate == Tachycardia then output health state = Unhealthy

RULE 7: IF Blood sugar == Prediabetic AND Heart rate == Bradycardia then output health state = Unhealthy

RULE 8: IF Blood sugar == Prediabetic AND Heart rate == Normal then output health state = Unhealthy

RULE 9: IF Blood sugar == Prediabetic AND Heart rate == Tachycardia then output health state = Unhealthy

RULE 10: IF Blood sugar == Diabetic AND Heart rate == Bradycardia then output health state = Unhealthy

RULE 11: IF Blood sugar == Diabetic AND Heart rate == Normal then output health state = Unhealthy

RULE 12: IF Blood sugar == Diabetic AND Heart rate == Tachycardia then output health state = Unhealthy

### 3.4. Outputs and Defuzzification

Output health state membership has two values: Unhealthy (<40) and Healthy (>60), as is shown in Figure 5. The output membership function limits are calculated as based on the calculations for the centroid method in [26]. The defuzzification function has the following characteristics:$$\mathrm{Unhealthy}=\left\{\begin{array}{c} 1;\;\;\;\;\;x\leq 40\\ \frac{60\;-\;x}{60\;-\;40};\;\;\;\;\;\;40<x<60\\ 0;\;\;\;\;\;\;x\geq 60 \end{array}\right.$$
$$\mathrm{Healthy}=\left\{\begin{array}{c} 1;\;\;\;\;x\geq 60\\ \frac{x\;-\;40}{60\;-\;40};\;\;\;\;40<x<60\\ \;\;\;\;\;\;0;\;\;\;\;\;x\leq 6=40 \end{array}\right..$$

### 3.5. Example

Let us use an example to understand the manual calculation. Consider a blood sugar value of 5.42 mmol/L and a heart rate of 58.7 BPM.

#### 3.5.1. Fuzzification

From the blood sugar membership function, the value of 5.42 mmol/L is in the normal and prediabetic intersection region. So, using membership equations, the fuzzy value can be calculated as follows:

Blood Sugar:$$\mathrm{Normal}=\frac{5.52\;-\;5.42}{5.5\;-\;5.4}=0.8$$
$$\mathrm{Prediabetic}=\frac{5.42\;-\;5.4}{5.5\;-\;5.4}=0.2$$

Similarly, the heart rate value of 58.7 BPM can be found in the heart rate membership function of the Bradycardia and Normal intersection region. Thus, using the corresponding equation of these ranges, the fuzzy value is determined as follows:

Heart rate:$$\mathrm{Bradycardic}=\frac{60\;-\;58.7}{60\;-\;58}=0.65$$
$$\mathrm{Normal}=\frac{58.7\;-\;58}{60\;-\;58}=0.35$$

#### 3.5.2. Rules

The value of the membership function is determined on the basis of the assigned rules as given in Table 2 and Figure 6. Since each rule uses an AND operation, the minimum value between the two is known to be the output value. Applying the above measured value to the rules we are given the following inference:

RULE 4: IF Blood sugar == Normal AND Heart rate == Bradycardia then output health state = Unhealthy IF Blood sugar == 0.8 AND Heart rate == 0.65 then output = 0.65 Unhealthy

RULE 5: IF Blood sugar == Normal AND Heart rate == Normal then output health state = Healthy IF Blood sugar == 0.8 AND Heart rate == 0.35 then output = 0.35 Healthy

RULE 7: IF Blood sugar == Prediabetic AND Heart rate == Bradycardia then output health state = Unhealthy IF Blood sugar == 0.2 AND Heart rate == 0.65 then output = 0.2 Unhealthy

RULE 8: IF Blood sugar == Prediabetic AND Heart rate == Normal then output health state = Unhealthy IF Blood sugar == 0.2 AND Heart rate == 0.35 then output = 0.2 Unhealthy

#### 3.5.3. Defuzzification

To determine if the person is healthy or unwell, the output membership function is taken into account for defuzzification. The values of the membership functions are obtained from the rules defined using fuzzy logic. Now, we measure the defuzzification value using the centroid method [9,26].
$$\mathrm{Output}\;\mathrm{status}=\frac{0.65*40+0.35*60+0.2*40+0.2*40}{0.65+0.35+0.2+0.2}=45$$

The calculated output value can be plotted using the output membership function. The output value of 45 lies between healthy and unhealthy in their intersection region. It is therefore concluded that the patient has not fully fallen into an unhealthy state. The patient does not need to take any medicine as prescribed for an obese person. By lifestyle change and appropriate workouts, he can return to a stable state.

### 3.6. Fuzzy Logic for Determining Insulin Dose

The first stage of a fuzzy expert system is to diagnose whether a person is healthy or unhealthy. This output is considered to be one of the inputs to the second stage of the fuzzy expert system used to determine the dose of insulin for the diabetic patient [28]. Four input functions, including health status output from the previous fuzzy step, diabetic stage, patient weight, and carbohydrate intake, are considered in this Insulin Dose Assistive Fuzzy Process as shown in Figure 7.

For the first input membership function, the health state which is used for fuzzification is obtained from the previous fuzzy system. The membership function for the health state is shown in Figure 8. The membership function discusses two health states: Unhealthy (<40) and Healthy (>60).
$$\mathrm{Unhealthy}=\left\{\begin{array}{c} 1;\;\;y\leq 40\\ \frac{60\;-\;y}{60\;-\;40};\;\;40<y<60\\ 0;\;\;\;y\geq 60 \end{array}\right.$$
$$\mathrm{Healthy}=\left\{\begin{array}{c} 1;\;\;y\geq 60\\ \frac{y\;-\;40}{60\;-\;40};\;\;40<y<60\\ 0;\;\;\;y\leq 40 \end{array}\right.$$

The next input membership function is the diabetic level, which is determined using the blood glucose level of the patient. The membership function is considered to have three levels of diabetic condition: Low (7.4 mmol/L to 8 mmol/L), Medium (8.4 mmol/L to 8.8 mmol/L) and High (>9.1 mmol/L) as shown in Figure 9.
$$\mathrm{Low}=\left\{\begin{array}{cc} 1,\;\;\;\;\;\;\;\;\;\;\; & 7.4\leq y\leq 8\\ \frac{y\;-\;7.0}{7.4\;-\;7.0},\;\;\;\;\;\;\;\;\;\;\; & 7.0<y<7.4\\ \frac{8.4\;-\;y}{8.4\;-\;8.0},\;\;\;\;\;\;\;\;\;\;\; & 8.0<y<8.4\;\\ 0,\;\;\;\;\;\;\;\;\;\;\; & y\leq 7.0;\;y\geq 8.4 \end{array}\right.$$
$$\mathrm{Medium}=\left\{\begin{array}{cc} 1,\;\;\;\;\;\;\;\;\;\;\; & 7.4\leq y\leq 8\\ \frac{y\;-\;7.0}{7.4\;-\;7.0},\;\;\;\;\;\;\;\;\;\;\; & 7.0<y<7.4\\ \frac{8.4\;-\;y}{8.4\;-\;8.0},\;\;\;\;\;\;\;\;\;\;\; & 8.0<y<8.4\;\\ 0,\;\;\;\;\;\;\;\;\;\;\; & y\leq 7.0;\;y\geq 8.4 \end{array}\right.$$
$$\mathrm{High}=\left\{\begin{array}{c} 1;\;\;y\geq 9.1\\ \frac{y\;-\;8.8}{9.1\;-\;8.8};\;\;8.8<y<9.1\\ 0;\;\;\;y\leq 8.8 \end{array}\right.\;$$

In order to determine the dose value of insulin, the patient’s weight must be taken into account. Obesity and idle lifestyle are usually two factors that can lead to many health problems, including diabetic conditions. Therefore, in this fuzzy logic system, the weight of the patient is considered to be one of the input parameters. Weight is divided into three ranges: low (50 kg to 65 kg), moderate (70 kg to 75 kg) and high (80 kg to 95 kg). The membership function for the person’s weight is shown in Figure 10 and is defined as follows:
$$\mathrm{Low}=\left\{\begin{array}{c} 1;\;\;y\geq 65\\ \frac{70\;-\;y}{70\;-\;65};\;\;\;65<y<70\\ 0;\;\;\;y\geq 70 \end{array}\right.$$
$$\mathrm{Moderate}=\left\{\begin{array}{cc} 1,\;\;\;\;\;\;\;\;\;\;\; & 70\leq y\leq 75\\ \frac{y\;-\;65}{70\;-\;65},\;\;\;\;\;\;\;\;\;\;\; & 65<y<70\\ \frac{80\;-\;y}{80\;-\;75},\;\;\;\;\;\;\;\;\;\;\; & 75<y<80\\ 0,\;\;\;\;\;\;\;\;\;\;\; & y\leq 65;\;y\geq 80 \end{array}\right.$$
$$\mathrm{High}=\left\{\begin{array}{cc} 1,\;\;\;\;\;\;\;\;\;\;\; & 80\leq y\leq 95\\ \frac{y\;-\;75}{80\;-\;75},\;\;\;\;\;\;\;\;\;\;\; & 75<y<80\\ \frac{100\;-\;y}{100\;-\;95},\;\;\;\;\;\;\;\;\;\;\; & 95<y<100\;\\ 0,\;\;\;\;\;\;\;\;\;\;\; & y\leq 75;\;y\geq 100 \end{array}\right.\;$$

Carbohydrates are the foundation of a healthy diet. However, carbs have an effect on a person’s blood sugar level, so it is crucial to control carbohydrate consumption. The carbohydrate intake level of the patient should also be treated as an input parameter in order to provide an assist tool for calculating the dose of insulin. The carbohydrate membership function is shown in Figure 11. The range of carbohydrates is divided as: low (<235 g), medium (240–275 g) and large (280–355 g).
$$\mathrm{Low}=\left\{\begin{array}{c} 1;\;\;y\leq 235\\ \frac{240\;-\;y}{240\;-\;235};\;\;235<y<240\\ 0;\;\;\;y\geq 240 \end{array}\right.$$
$$\mathrm{Medium}=\left\{\begin{array}{cc} 1,\;\;\;\;\;\;\;\;\;\;\; & 240\leq y\leq 275\\ \frac{y\;-\;235}{240\;-\;235},\;\;\;\;\;\;\;\;\;\;\; & 235<y<240\\ \frac{280\;-\;y}{280\;-\;275},\;\;\;\;\;\;\;\;\;\;\; & 275<y<280\\ 0,\;\;\;\;\;\;\;\;\;\;\; & y\leq 235;\;y\geq 280 \end{array}\right.$$
$$\mathrm{High}=\left\{\begin{array}{cc} 1,\;\;\;\;\;\;\;\;\;\;\; & 280\leq y\leq 355\\ \frac{y\;-\;275}{280\;-\;275},\;\;\;\;\;\;\;\;\;\;\; & 275<y<280\\ \frac{360\;-\;y}{360\;-\;355},\;\;\;\;\;\;\;\;\;\;\; & 355<y<360\;\\ 0,\;\;\;\;\;\;\;\;\;\;\; & y\leq 275;\;y\geq 360 \end{array}\right.$$

### 3.7. Rules for Fuzzy Logic

According to studies conducted by the authors of [29], almost 40 percent of the total daily insulin dose is used to replace insulin during fasting or between meals. The remaining 60 percent is used to correct the blood sugar and carbohydrate coverage. The total dose of insulin is determined based on patient weight, daily intake of carbohydrate and blood sugar.

The overall requirement for daily insulin is measured as tolerable daily intake (TDI (in insulin units)) = 0.55 × the total weight of a kilogram. Based on the measured TDI, the carbohydrate coverage ratio (CCR) can be calculated as CCR = 500/TDI. The dose of carbohydrate (CHO) insulin is calculated by taking the ratio of the daily carbohydrate intake in grams to CCR. Blood sugar is decreased by 50 mg/dL by 1 unit dose and thus the high blood sugar correction factor is 50. It is a fact that pre-meal blood sugar is 120 mg/dL under normal conditions. Therefore, in order to calculate the correct dose of CHO insulin measured, the formula is a cumulative dose (CD) = (Blood sugar level in mg/dL − 120) × TDI/1800. The final accumulated dose of insulin is then obtained by adding the CHO insulin dose to the CD correction dose. Based on these calculations performed on various input range values, the rules for the measurement of insulin dosage for fuzzy logic are summarized in Table 3.

### 3.8. Outputs and Defuzzification

The output insulin dose membership has four insulin dose-dependent levels as shown in Figure 12: low (<20 units), medium (22 to 28 units), moderate (30 to 36 units), very moderate (38 to 42 units).
$$\mathrm{Low}=\left\{\begin{array}{c} 1;\;\;y\leq 20\\ \frac{22\;-\;y}{22\;-\;20};\;\;20<y<22\\ 0;\;\;\;y\geq 22 \end{array}\right.$$
$$\mathrm{Medium}=\left\{\begin{array}{cc} 1,\;\;\;\;\;\;\;\;\;\;\; & 22\leq y\leq 28\\ \frac{y\;-\;20}{22\;-\;20},\;\;\;\;\;\;\;\;\;\;\; & 20<y<22\\ \frac{30\;-\;y}{30\;-\;28},\;\;\;\;\;\;\;\;\;\;\; & 28<y<30\;\\ 0,\;\;\;\;\;\;\;\;\;\;\; & y\leq 20;\;y\geq 30 \end{array}\right.$$
$$\mathrm{High}=\left\{\begin{array}{cc} 1,\;\;\;\;\;\;\;\;\;\;\; & 30\leq y\leq 36\\ \frac{y\;-\;28}{30\;-\;28},\;\;\;\;\;\;\;\;\;\;\; & 28<y<30\\ \frac{38\;-\;y}{38\;-\;36},\;\;\;\;\;\;\;\;\;\;\; & 36<y<38\;\\ 0,\;\;\;\;\;\;\;\;\;\;\; & y\leq 28;\;y\geq 38 \end{array}\right.$$
$$\mathrm{Very}\;\mathrm{High}=\left\{\begin{array}{cc} 1,\;\;\;\;\;\;\;\;\;\;\; & 38\leq y\leq 42\\ \frac{y\;-\;36}{38\;-\;36},\;\;\;\;\;\;\;\;\;\;\; & 36<y<38\\ \frac{44\;-\;y}{44\;-\;42},\;\;\;\;\;\;\;\;\;\;\; & 42<y<44\;\\ 0,\;\;\;\;\;\;\;\;\;\;\; & y\leq 36;\;y\geq 44 \end{array}\right.$$

### 3.9. Example

Let us use an example in order to understand the entire manual calculation for fuzzification and defuzzification of insulin dosage calculation. Consider the health state value of 45, the diabetic blood sugar level 8.2 mmol/L, weight of the patient 68 kg and the carbohydrate intake 238 g.

#### 3.9.1. Fuzzification

From the health state membership input function, the value 45 is in the healthy and unhealthy intersection region. So, using membership equations, the fuzzy value can be calculated as follows:

Health State:$$\mathrm{Unhealthy}=\frac{60\;-\;45}{60\;-\;40}=0.75$$
$$\mathrm{Healthy}=\frac{45\;-\;40}{60\;-\;40}=0.25$$

From the diabetic blood sugar membership function, the value 8.2 mmol/L is in the Low and Medium intersection region. So, using membership equations, the fuzzy value can be calculated as follows:

Diabetic Blood Sugar Level:$$\mathrm{Low}=\frac{8.2\;-\;7.0}{7.4\;-\;7.0}=3$$
$$\mathrm{Medium}=\frac{8.2\;-\;8.0}{8.4\;-\;8.0}=0.5$$

From the patient’s weight membership function, the value 68 kg is in the Low and Moderate intersection region. So, using membership equations, the fuzzy value can be calculated as follows:

Weight:$$\mathrm{Low}=\frac{70\;-\;68}{70\;-\;65}=0.4$$
$$\mathrm{Moderate}=\frac{68\;-\;65}{70\;-\;65}=0.6$$

Similarly, the daily carbohydrate intake value of 238 g can be found in the membership function in the Low and Medium intersection region. Thus, using the corresponding equation of these ranges, the fuzzy value is determined as follows:

Carbohydrate Intake:$$\mathrm{Low}=\frac{240\;-\;238}{240\;-\;235}=0.4$$
$$\mathrm{Normal}=\frac{238\;-\;235}{240\;-\;235}=0.6$$

#### 3.9.2. Rules

The value of the membership function is determined on the basis of the assigned rules in Table III. Since each rule uses an AND operation, the minimum value among the four is considered to be the output value. Applying the above measured value to the rules we are given the following inference:

RULE 1: 

IF Health State == Unhealthy AND Diabetic Level == Low AND Weight == Low AND Carbohydrate Intake == Low, then output Insulin Dose = Low

IF Health State == 0.75 AND Diabetic Level == 3 AND Weight == 0.4 AND Carbohydrate Intake == 0.4, then output Insulin Dose = 0.4 Low

RULE 2:

IF Health State == Unhealthy AND Diabetic Level == Low AND Weight == Low AND Carbohydrate Intake == Medium, then output Insulin Dose = 0.4 Low

IF Health State == 0.75 AND Diabetic Level == 3 AND Weight == 0.4 AND Carbohydrate Intake == 0.4, then output Insulin Dose = 0.4 Low

RULE 4:

IF Health State == Unhealthy AND Diabetic Level == Low AND Weight == Moderate AND Carbohydrate Intake == Low, then output Insulin Dose = Low

IF Health State == 0.75 AND Diabetic Level == 3 AND Weight == 0.6 AND Carbohydrate Intake == 0.4, then output Insulin Dose = 0.4 Low

RULE 5:

IF Health State == Unhealthy AND Diabetic Level == Low AND Weight == Moderate AND Carbohydrate Intake == Medium, then output Insulin Dose = Medium

IF Health State == 0.75 AND Diabetic Level == 3 AND Weight == 0.6 AND Carbohydrate Intake == 0.6, then output Insulin Dose = 0.6 Medium

RULE 10:

IF Health State == Unhealthy AND Diabetic Level == Medium AND Weight == Low AND Carbohydrate Intake == Low, then output Insulin Dose = Low

IF Health State == 0.75 AND Diabetic Level == 0.5 AND Weight == 0.4 AND Carbohydrate Intake == 0.4, then output Insulin Dose = 0.4 Low

RULE 11:

IF Health State == Unhealthy AND Diabetic Level == Medium AND Weight == Low AND Carbohydrate Intake == Medium, then output Insulin Dose = Low

IF Health State == 0.75 AND Diabetic Level == 0.5 AND Weight == 0.4 AND Carbohydrate Intake == 0.6, then output Insulin Dose = 0.4 Low

RULE 13:

IF Health State == Unhealthy AND Diabetic Level == Medium AND Weight == Moderate AND Carbohydrate Intake

== Low, then output Insulin Dose = Low

IF Health State == 0.75 AND Diabetic Level == 0.5 AND Weight == 0.6 AND Carbohydrate Intake == 0.4, then output Insulin Dose = 0.4 Low

RULE 14:

IF Health State == Unhealthy AND Diabetic Level == Medium AND Weight == Moderate AND Carbohydrate Intake

== Medium, then output Insulin Dose = Medium

IF Health State == 0.75 AND Diabetic Level == 0.5 AND Weight == 0.6 AND Carbohydrate Intake == 0.6, then output Insulin Dose = 0.5 Medium

#### 3.9.3. Defuzzification

For defuzzification, the insulin dose output membership function is considered to infer the insulin dosage needed for the patient according to the values of the input membership functions, health status, diabetic level, patient weight, and intake of carbohydrates. The values of the membership functions are derived from the rules specified by fuzzy logic as shown in Table 3. By using the centroid method [9,26], we can calculate the defuzzification value as follows
$$\mathrm{Output}\;\mathrm{status}=\frac{0.4*20+0.4*20+0.4*20+0.6*22+0.4*20+0.4*20+0.4*20+0.5*28}{0.4+0.4+0.4+0.6+0.4+0.4+0.4+0.5}=18.34\;\mathrm{units}$$

## 4. Experimental Setup

The fuzzy logic problem described in the previous section is simulated using MATLAB. For the implementation, the fuzzy logic tool is used. The fuzzy logic system to diagnose diabetics is designed as a two-input and a single output system. The inputs are Blood Sugar and Heart Rate. The ranges for these two inputs are specified in a fuzzy logic tool similar to those of the ranges defined in the previous sections. The output of the fuzzy logic system is a state of health; healthy or unhealthy. The rules for fuzzy logic are programmed as shown in Figure 6. The output health status of any set of input values is calculated by the fuzzy logic method in compliance with the fuzzy rules. The inputs and outputs for the fuzzy system to diagnose a diabetic can be summarized as given in Table 4:

The next stage of the fuzzy logic system is to calculate the insulin dosage for the diabetic patients, which has four input membership functions and one output. The inputs are the health state obtained from the previous state, diabetic level, weight of the patient and the daily carbohydrate intake. The rules are set in the fuzzy logic system as shown in Table 3. The output membership function for the fuzzy system is the insulin dosage levels. The parameters for the fuzzy system for the insulin dosage calculation can be summarized as given in Table 5.

## 5. Simulation Result

The fuzzy logic toolbox in MATLAB provides a solution mechanism for any problems highlighted by the input and output set. Therefore, we have chosen MATLAB as the simulation platform for this work. The results of the simulation are shown in Figure 13, which indicates the patient’s health status with the chosen input combination of heart rate and blood sugar. According to the membership function of blood sugar in the fuzzy system used for diabetic diagnosis, the patient’s blood sugar is considered normal if it is between 4.0 mmol/L and 5.4 mmol/L. According to the membership function of heart rate, a normal heart rate is one that ranges between 60 and 100 beats per minute. In Figure 13, the blood sugar and heart rate values are within normal ranges. According to the rules in Table 2, if the heart rate and blood sugar levels are normal, the output health state is also normal. The patient’s health status is considered healthy if the centroid method gives a result greater than 60. The output value in Figure 13 is within the correct range, indicating that the patient is in a healthy output state. Furthermore, the surface graph in Figure 14 shows whether the patient is healthy or unhealthy for the given input combination of heart rate and blood sugar. The fuzzy system for diabetes diagnosis follows the guidelines in Table 2. The patient is healthy only when the blood sugar and heart rate values are normal. In all other cases, the patient’s health is deemed unhealthy. Therefore, when the blood sugar and heart rate are within normal ranges, the surface viewer for health state diagnosis indicates that the health state value is on the high side (healthy). The graph clearly shows that when the patient’s blood sugar level is in the normal range (4.0 mmol/L to 5.4 mmol/L), he or she is in a healthy state, resulting in a centroid method value of 60 or higher. All other lower values indicate that the patient is pre-diabetic or diabetic, and the blood sugar ranges considered in the blood sugar membership function indicate that the patient’s condition is unhealthy.

Figure 15 shows the output plot obtained when only the input value of blood sugar is given, indicating the range of input values of blood sugar considered for simulation. As per figure, it clearly indicates that when the blood sugar value is in the range of a normal condition (4.0 mmol/L to 5.4 mmol/L), then the patient is in a perfect healthy output state which is considered to be 60 above in centroid method. In all other values of low, pre-diabetic or diabetic range considered in the membership function of blood sugar, the patient condition ranges towards the unhealthy output state. Figure 16 shows the output plot with only the heart rate input. According to the membership function of heart rate, it is normal if the heart rate ranges between 60 and 100 beats per minute. During this range, a patient is in a healthy output state. The output plot of health state can be seen ranging towards the unhealthy output state in all other heart rate conditions (Bradycardia and Tachycardia).

The fuzzy logic system for insulin calculation considers four inputs: the person’s health, diabetes level, weight, and carbohydrate intake. According to the input values in Figure 17, the health state is unhealthy (26.5), the diabetic level is medium (8.65 mmol/L), the weight is moderate (71.6 kg), and the carbohydrate intake is medium (260 g). According to the rules in Table 3, the centroid method of calculation indicates that the calculated insulin dosage is also in the medium range (27.5). Figure 18 depicts a 3 D view of all potential output insulin doses for various diabetic and carbohydrate intake amounts. Figure 19 depicts a surface view of the output dose based on the patient’s weight and diabetic status. Figure 18 and Figure 19 show that the calculated insulin dosage is in the high range if the diabetic level, body weight, and carbohydrate intake are all in the high or very high range. When these parameters fall within the low or moderate levels, insulin dosages also fall within these limits.

To compare the outcomes with the simulated output value for the insulin dosage, calculations using real-time values have been carried out. For input blood sugar level 8.65 mmol/L, weight 71.6 kg, daily carbohydrate intake 260 g, the total daily insulin requirement is 39.38, the CHO insulin dose is 20.48 and correction dose is 7.25. Thus, the final insulin dosage is calculated as follows: CHO insulin dose + Correction dose = 27.73 units. The simulated output utilizing fuzzy logic is calculated as 27.5 units as from Figure 17. In order to determine that the simulated fuzzy output can be approximated to the calculated value, the results are compared.

## 6. Conclusions

This work develops an expert remote care system for diagnosing and treating diabetic patients by assisting them in determining the appropriate insulin dosage. The proposed system is an assisted tool for daily insulin dosage adjustment for diabetic patients based on their daily blood sugar value, carbohydrate intake and their body weight. This is a prototype that has not undergone commercial testing. This system operates in two stages, the first of which takes the glucose level and the patient’s heart rate as input into the fuzzy system. Based on the input, the fuzzy system determines whether or not the patient is diabetic. The next round of treatment will commence once the patient is diagnosed with diabetes. For this purpose, another fuzzy system is used which takes into account the glucose level, age and calorie intake as input by the patient. This determines the appropriate dosage. This type of system is beneficial to both the patient and the hospital since it reduces the frequency of hospital visits, monitors the patient often, and lowers the cost of care. In terms of serving a large number of patients and offering services to a wide range of patients who also live in distinct cities, the hospital and the doctor both benefit. As a concluding point, this method is more important than ever, especially in the midst of a pandemic like COVID, when hospital facilities and medical personnel are fully occupied and chronically ill individuals may be overlooked.

## Figures and Tables

**Figure 1 healthcare-11-00012-f001:**
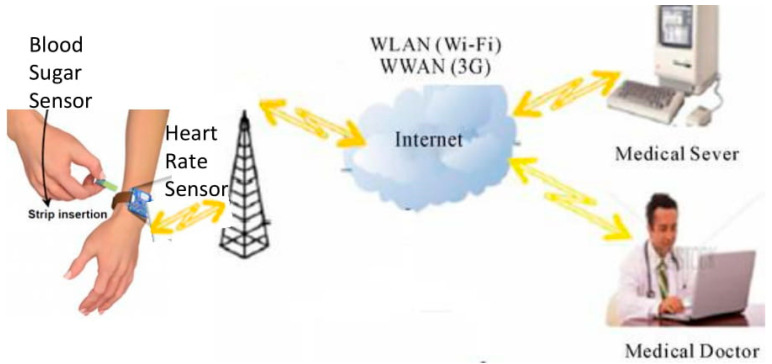
System Overview.

**Figure 2 healthcare-11-00012-f002:**
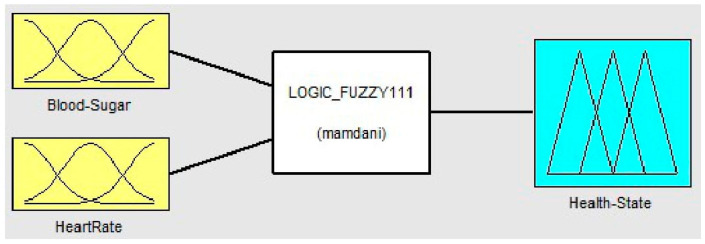
Overview of Fuzzy Logic System.

**Figure 3 healthcare-11-00012-f003:**
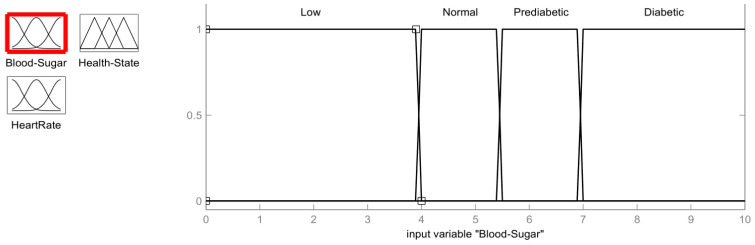
Membership function of Blood Sugar in mmol/L.

**Figure 4 healthcare-11-00012-f004:**
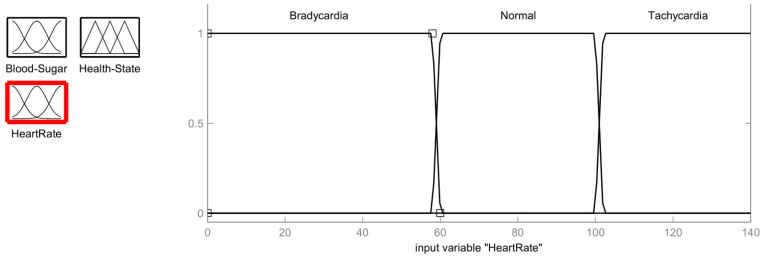
Membership function of Heart Rate in BPM.

**Figure 5 healthcare-11-00012-f005:**
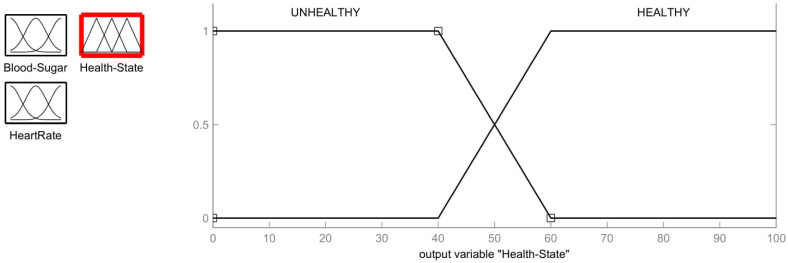
Membership function for Output Health State.

**Figure 6 healthcare-11-00012-f006:**
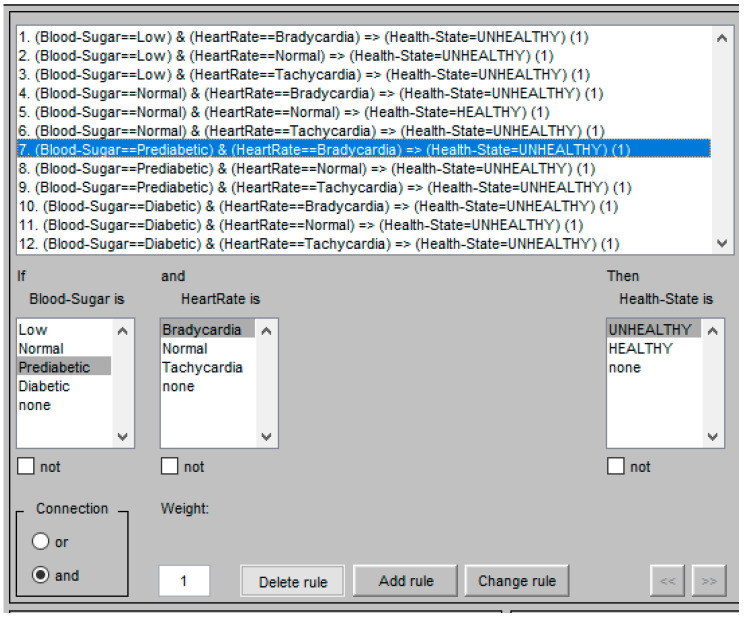
Rules in Matlab Fuzzy Logic Tool to determine the health state.

**Figure 7 healthcare-11-00012-f007:**
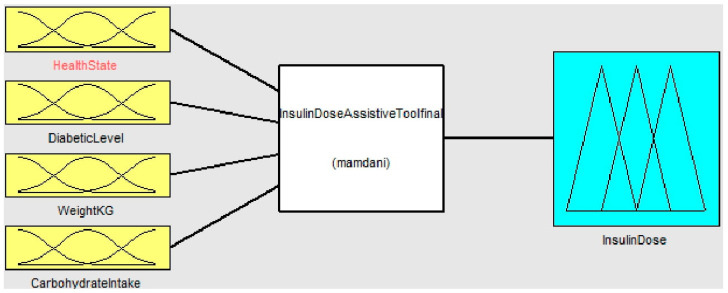
Insulin Dose Assistive Tool Overview.

**Figure 8 healthcare-11-00012-f008:**
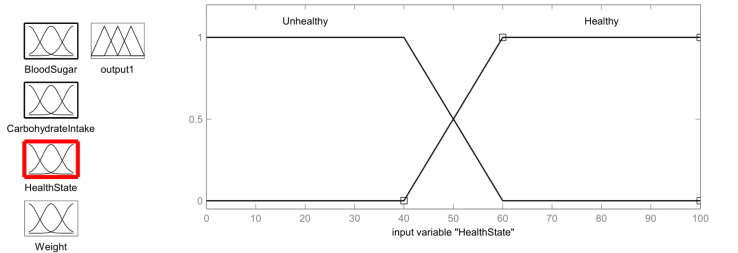
Health State Membership function.

**Figure 9 healthcare-11-00012-f009:**
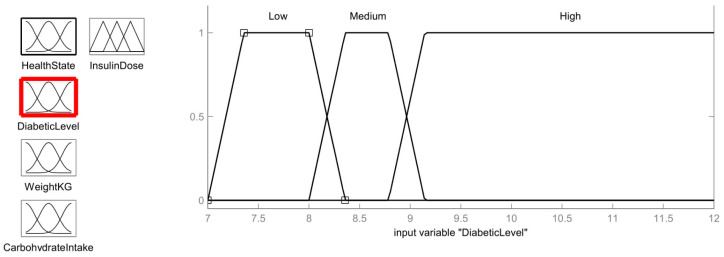
Diabetic Level Membership function.

**Figure 10 healthcare-11-00012-f010:**
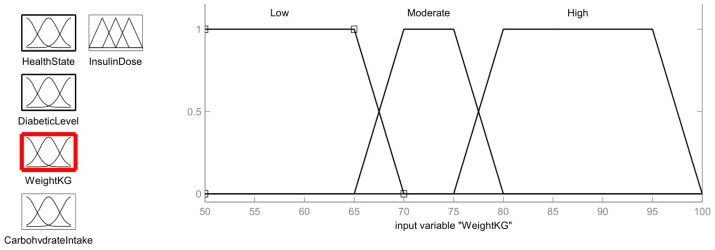
Patient’s Weight Membership function.

**Figure 11 healthcare-11-00012-f011:**
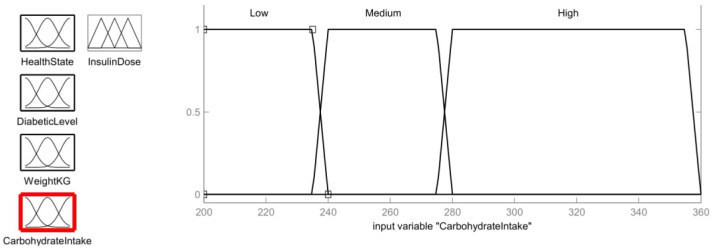
Carbohydrate Intake Membership function.

**Figure 12 healthcare-11-00012-f012:**
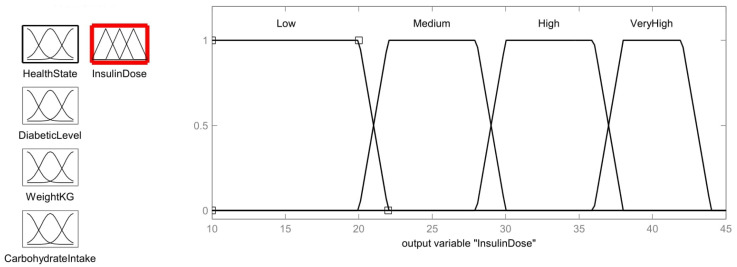
Insulin dose Membership function.

**Figure 13 healthcare-11-00012-f013:**
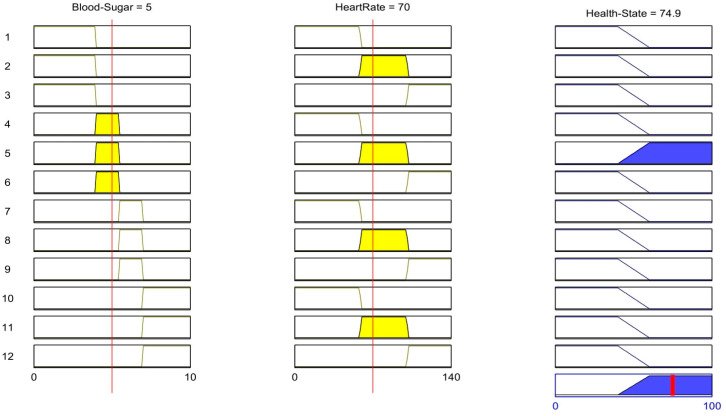
Output Plot.

**Figure 14 healthcare-11-00012-f014:**
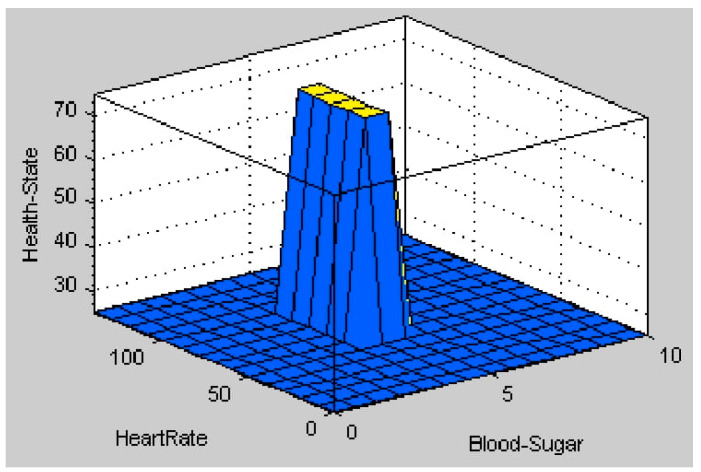
Surface Viewer for Health State diagnosis.

**Figure 15 healthcare-11-00012-f015:**
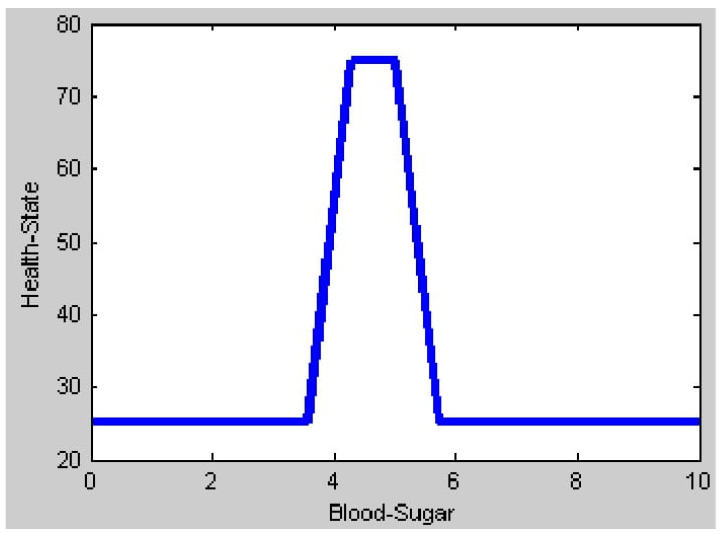
Output plot for Blood Sugar input.

**Figure 16 healthcare-11-00012-f016:**
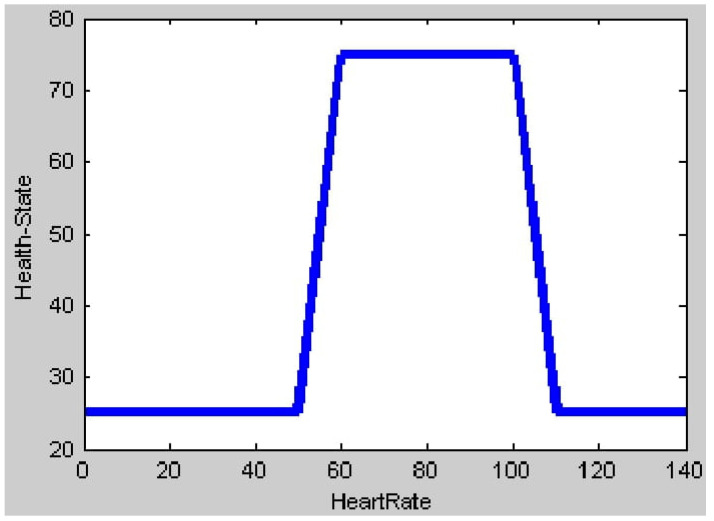
Output plot for Heart rate input.

**Figure 17 healthcare-11-00012-f017:**
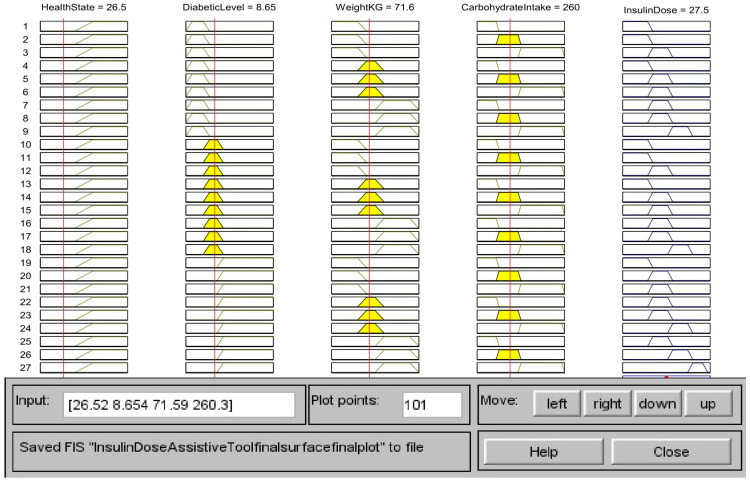
Output Insulin calculation based on the rules.

**Figure 18 healthcare-11-00012-f018:**
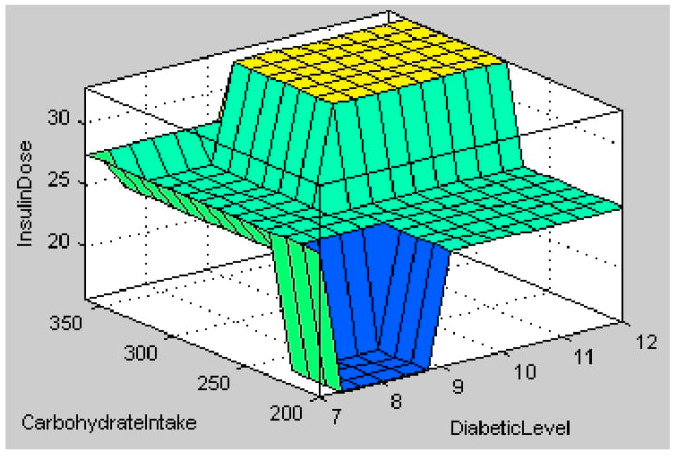
Surface view output for Inputs Diabetic Level and Carbohydrate Intake.

**Figure 19 healthcare-11-00012-f019:**
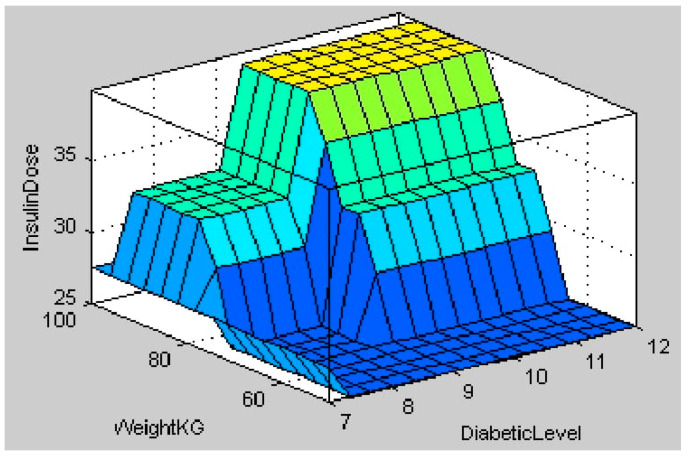
Surface view output for Inputs Diabetic Level and Weight.

**Table 1 healthcare-11-00012-t001:** Inputs for fuzzification.

Blood Sugar	Low, Normal, Prediabetic, Diabetic
Heart Rate	Bradycardia, Normal, Tachycardia

**Table 2 healthcare-11-00012-t002:** Rules for Fuzzification.

Blood	Heart Rate		
Sugar	Bradycardia	Normal	Tachycardia
Low	Unhealthy	Unhealthy	Unhealthy
Normal	Unhealthy	Healthy	Unhealthy
Prediabetic	Unhealthy	Unhealthy	Unhealthy
Diabetic	Unhealthy	Unhealthy	Unhealthy

**Table 3 healthcare-11-00012-t003:** Rules for Fuzzification.

Rules	Diabetic	Weight	Carbohydrate	Insulin
1	Low	Low	Low	Low
2	Low	Low	Medium	Low
3	Low	Low	High	Medium
4	Low	Moderate	Low	Low
5	Low	Moderate	Medium	Medium
6	Low	Moderate	High	Medium
7	Low	High	Low	Medium
8	Low	High	Medium	Medium
9	Low	High	High	High
10	Medium	Low	Low	Low
11	Medium	Low	Medium	Low
12	Medium	Low	High	Medium
13	Medium	Moderate	Low	Low
14	Medium	Moderate	Medium	Medium
15	Medium	Moderate	High	Medium
16	Medium	High	Low	Medium
17	Medium	High	Medium	Medium
18	Medium	High	High	High
19	High	Low	Low	Low
20	High	Low	Medium	Medium
21	High	Low	High	Medium
22	High	Moderate	Low	Medium
23	High	Moderate	Medium	Medium
24	High	Moderate	High	High
25	High	High	Low	Medium
26	High	High	Medium	High
27	High	High	High	Very High

**Table 4 healthcare-11-00012-t004:** Parameters for fuzzy system to diagnose diabetic.

	Parameter	Membership Function			
Input	Blood Sugar (mmol/L)	Low (0–3.9)	Normal (4.0–5.4)	Prediabetic (5.5–6.9)	Diabetic (>7.0)
Input	Heart Rate (BPM)	Bradycardia (<58)	Normal (60–100)	Tachycardia (>102)	-
Output	Health State	Unhealthy (<40)	Healthy (>60)	-	-

**Table 5 healthcare-11-00012-t005:** Parameters for fuzzy system for insulin dosage.

	Parameter	Membership Function			
Input	Health State	Unhealthy (<40)	Healthy (>60)	-	-
Input	Diabetic Level (mmol/L)	Low (7.4–8.0)	Medium (8.4–8.8)	High (>9.1)	-
Input	Weight (kg)	Low (50–65)	Moderate (70–75)	High (80–95)	-
Input	Carbohydrate intake (g)	Low (<235)	Medium (240–275)	High (30–36)	-
Output	Insulin Dose (Units)	Low (<20)	Medium (22–28)	High (280–355)	Very High (38–42)

## Data Availability

Not applicable.

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
