# Peer review of "IoT Based Expert System for Diabetes Diagnosis and Insulin Dosage Calculation"

_healthcare, 2022, doi:10.3390/healthcare11010012_

Round 1

Reviewer 1 Report

I think it’s interesting research. I would recommend it if the following issues are handled properly

1. Please provide a detailed description of your IoT devices.

2. Please explain the algorithms of your expert system and highlight the advantages of that.

3. Improve the Figure's quality and presentation modes based on some published papers.

4. Highlight the novelties in this research and their clinical significance.

5. Add some discussion about the accuracy and reliability of this expert system for diabetes diagnosis and insulin dosage calculation.

Author Response

The response is attached.

Reviewer 2 Report

Summary:

After a comprehensive motivation for remote patient monitoring (RPM) and its application to diabetes in particular, the proposed system is presented, consisting of two major parts: Wearable sensors and Smartphone at the patient’s side for monitoring and a clinic-based backend. The parts are connected using established technology like Bluetooth- and internet-connections. The decision-making and insulin-dose recommendations are determined based on a fuzzy logic-based approach.

Subsequently, the related work is presented and the novelty of the approach presented in this contribution explained. Afterwards, the proposed system explained in detail.

The formulas for fuzzification and defuzzification are presented in detail and the sound generation of outputs using the fuzzy-logic approach are shown. Based on simulation date, the proper function of the calculations is shown.
An evaluation showing the usability and benefits of this system for patient and expert in a real patient-situation is missing.

Comments:

The section 2 on related work misses some references for some more general statements that sound reasonable, but are not proven by references. However, there is space to mention the author’s own publication. E.g.: Lines 131 through 136. Elaboration on other academically proposed approaches seems quite

Though the process of fuzzification and defuzzification is explained in detail, even using am example, the specific values chosen for the rules for (de)fuzzification are just presented, but not argued for or supported by references. The layout and use of abbreviations, formulas and units are not suitable for a well readable text.

Table 1 appears to have a broken layout; relevant information is cut-off. Table 2 seems to have a layout-issue as well.

Is it necessary to present the rules for fuzzification twice, in a table and in 12 written formulas?

The layout of tables, formulas and fuzzy-rules shall be improved to increase readability.

Section 3.7 suffers from use of un-introduced abbreviations, unclear notation in formulas and unprecise use of units. This section shall be revised wrt. the presentation of the used formulas and units.

The design of the system and the used formulas are explained in detail, which is clearly a highlight of the contribution, but lack a proper presentation e.g., wrt layout.

Though elaborated on in the section on related work, the subsequent text does not explain anything about the implementation of the system at the patient nor about the expert-system in the clinic. It becomes clear to the end of section 3 that the paper is only presenting the design of the underlying software system, but does not take into account any of the implementation-related aspects such as reliability, usability (for patient and expert) or security. It is said that the system uses Bluetooth-connections and smartphones at the patient, as well as internet-communication and some server-infrastructure at the clinic. These are components that need to be designed and operated with security-aspects considered, which will have impact on their use. Despite the outlook in the abstract and the section on related work, these implementation- and usage-related aspects are missing.
I suggest to be more clear on the focus of the text from the beginning. This includes the clarification that the evaluation is done purely based on simulated data.  

The evaluation is rather a software test to proof that the system is working properly. There is no data showing that the system is generating responses that indicate its usability in daily monitoring and treatment of diabetes. Among others, the comparison of the system’s output to the medication suggested by a human expert is missing in the discussion of the results.

In addition to the analysis of the system based on simulation, a comparison of the obtained decisions (diagnosis and prescription suggestion) to before mentioned related works that cover this aspect of the proposal, would allow a better rating of the accomplished.

Overall, an interesting contribution that is slightly fuzzy when it comes to the evaluation methodology and the discussion of the results. It shall be revised to gain more clarity on the focus of the paper and also wrt layout, punctuation and presentation of formulas etc. to reach an adequate level of readability.

Author Response

The response is attached.

Reviewer 3 Report

 The work presents interest for the diabetes diagnosis and insulin dosage calculation. The article consists in the presentation of an expert system development that can provide remote treatment for diabetic patients. The expert system is divided into two parts: one for the patient and the other for the hospital. The feedback is given to a healthy person after the first fuzzy process, while the diabetic patient receives feedback after the second fuzzy process.

The manuscript follows correctly the template imposed by the magazine.

In the introduction the authors justify the need of their research. The related work contains adequate documentation.

I do have some remarques:

1. The current version of this paper looks like an instruction manual of a self-developed software. For the experts in IT there’s not need so many explanations and for experts in medicine there is not necessary to write an instruction manual because they will not use it.

2. The paper didn’t present a real application, in a hospital, of the expert system.

The conclusions are too general. Elaborate more regarding added value and future work.

The references seem to be appropriate.

Author Response

The response is attached.

Reviewer 4 Report

This paper proposes an IoT based expert system for diabetes diagnosis and insulin dosage calculation. Fuzzy mathematic is adopted in the system design. The novelty of the idea is insignificant, and the reason for settings of key parameters in the system design is unclear. Simulations are taken to demonstrate how the idea works, which can not prove its effectiveness nor advantages to existing methods. It will be better if there is a real prototype system implemented and experiments on real dataset are taken. 

Author Response

The response is attached.

Round 2

Reviewer 3 Report

The article was improved.

Author Response

Thanks for the comments.

Reviewer 4 Report

The organization and description of the key idea of the paper can be improved to make it easier to follow.

Author Response

We have updated and revised the manuscript extensively to enhance the readability.
